# Impact of the COVID-19 pandemic on daily life and worry among mothers in Bhaktapur, Nepal

Suman Ranjitkar[1], Tor A. Strand[2,3]*, Manjeswori Ulak[1,2], Ingrid Kvestad[3,4], Merina Shrestha[1], Catherine Schwinger[2], Ram K. Chandyo[5], Laxman Shrestha[1], Mari Hysing[6]

1 Department of Pediatrics, Child Health Research Project, Institute of Medicine, Tribhuvan University, Kathmandu, Nepal, 2 Center for Intervention Science in Maternal and Child Health, Centre for International Health, University of Bergen, Bergen, Norway, 3 Department of Research, Innlandet Hospital Trust, Lillehammer, Norway, 4 Regional Centre for Child and Youth Mental Health and Child Welfare, NORCE Norwegian Research Centre, Bergen, Norway, 5 Department of Community Medicine, Kathmandu Medical College, Kathmandu, Nepal, 6 Department of Psychosocial Science, Faculty of Psychology, University of Bergen, Bergen, Norway

* tor.strand@uib.no

**Data Availability Statement:** Data described in the manuscript, code book, and analytic code will be made available upon request pending application and approval by the Nepal Health Research Council

## Abstract

The COVID-19 pandemic has affected many aspects of daily life worldwide, but the impact may be higher for impoverished populations. The main aim of this study is to describe the impact of the COVID-19 pandemic on different aspects of daily life in mothers in Nepal. We included 493 mothers of children aged 54–71 months participating in a randomized controlled trial on vitamin B12 supplementation. Mothers answered questions regarding the exposure and impact of the pandemic on their daily lives, and pandemic-related worries and sleep problems. We examined the extent to which worry, and sleep problems differed between mothers according to their exposure to COVID-19, socioeconomic status, and previous symptoms of depression. The mean age (SD) of the mothers was 32.3 (4.6) years and 54% had education below the secondary level. Of the mothers, 5.4% had either been exposed to someone who had tested positive or who had a family member with COVID-19. One-third of the participants responded that the pandemic had affected their economic situation, employment, and family life to a great deal. Both mothers and fathers with educational levels above 10 years or households with higher socioeconomic status had significantly higher average worry scores (maternal $p = 0.020$ and paternal $p = 0.005$). Mothers with a history of symptoms of depression had significantly more worry-related sleep problems during the pandemic ($p = 0.020$) than those without a history of depressive symptoms. Our study underlines the negative impact of the COVID-19 pandemic on diverse aspects of everyday life of mothers in Nepal.

## Introduction

The first infection with the SARS-CoV-2 virus was identified at the end of 2019 in Wuhan city, China [1], and it spread worldwide from February 2020 with a devastating impact on society.

(NHRC) and the Regional Committee for Medical and Health Research Ethics in Norway. Requests for data should be sent to the authors, by contacting NHRC (http://nhrc.gov.np), or by contacting the Department of Global Health and Primary Care at the University of Bergen (post@igs.uib.no), or contacting the Department of Pediatrics, Tribhuvan University Institute of Medicine, Kathmandu, Nepal (chrp2015@gmail. com).

**Funding:** This study was funded by grants from the Innlandet Hospital Trust, the Thrasher Research Fund (award # 11512), and the South-Eastern Norway Regional Health Authority (grant # 2012090). TAS reports funding from the South-Eastern Norway Regional Health Authority (grant # 2012090), MU from Thrasher Research Fund (award # 11512), and CS from the Research Council of Norway through a grant to Centre for Intervention Science in Maternal and Child Health (CISMAC) for conducting this research. The funders had no role in study design, data collection and analysis, decision to publish, or preparation of the manuscript.

**Competing interests:** The authors have declared that no competing interests exist.

In addition to the morbidity and mortality related to the infection, the restrictions imposed by governments to control the spread of the virus, such as lockdown, social distancing, isolation, and quarantine measures, have had negative impacts on the people's economy and mental health [2, 3]. While these negative consequences are global, populations in low-to-middle-income countries (LMIC) with limited social support and resources to respond to the pandemic may be especially vulnerable [4].

In Nepal, the first case of COVID infection was identified in January and the second case in March 2020. The government of Nepal declared a strict lockdown from the end of March 2020. The level of restrictions to control the spread of the virus varied until August [5]. The restrictions were related to impacts across many life-domains including economy, health service use and food security. Many Nepalese people lost their jobs as a consequence of the restrictions and lockdowns [6]. Mothers of infants were especially hard hit by the imposed restrictions, reflected in the increase in stillbirths and neonatal mortality during the pandemic [7], as well as a reduction in the utilization of health services (4) and the losses to production and supply chain of food resulting in food insecurity [8].

A pandemic also has considerable psychological impact such as increase in worry, fear, distress, and anxiety in the population [9, 10]. During the early months of the pandemic in Nepal, high rates of psychological distress such as restlessness, fearfulness, anxiety and worry, and sadness due to the COVID-19 outbreak was reported in two questionnaire-based studies [11, 12]. Worry is a strong driver for sleep problems, and an expected reduced sleep quality during the pandemic has also been reported [13]. Worry has been associated both directly with the risk of the disease but also indirectly with the restrictions due to the disease [13]. The studies on impact and worry are mainly limited to the first phase of the pandemic, and less is known about the impact of the pandemic over time and across life domains.

Worry and worry-related sleep problems may be more severe for people in high-risk groups. Underlying chronic conditions such as diabetes and cancer affect the severity of the COVID condition [14], and may give rise to increased worry and distress [15, 16]. It has been suggested that this pattern may be even more pronounced in countries with limited health care services [10]. People with depression may be another vulnerable group for psychological impact and sleep problems. This was confirmed among French adults with higher levels of fear and worse sleep quality among adults with depression than those with no or mild mental health-related symptoms [17].

Socioeconomic factors may have an impact on worries related to the COVID-19 [18]. The social gradient of mental health is well established, and it is thus expected that the negative consequences may be heightened for those with low SES [19, 20]. However, this may not be generalizable for the negative impact in Nepal as lower psychological impact of COVID-19 in people with low educational qualification compared to those with higher education was suggested in an online survey [21].

In August 2020, there were 39,460 confirmed cases in Nepal, which increased to 77,817 in September 2020, 170,743 in October and 274,143 when reaching February 2021. The number of deaths from COVID-19 was 404 in September 2020 (the start of our study) and slightly increased to 646 in October and 839 in November the same year. There was, however, a drastic decreasing trend from December with only 52 death cases in February 2021 [14, 22] when we completed our data collection.

The present study was conducted in the Bhaktapur municipality of Nepal. The municipality has a population of 83000 [23] and boarders to the eastern part of Kathmandu, the capital city of Nepal. It is a modern lower income setting characterized by mixed economy in an agriculture-based semi-urban community with many migrant workers from nearby districts. We collected information on COVID-19 exposure, impact on various aspects of daily life, worry for

infections, and worry-related sleep problems from September 2020 to February 2021in this district. The objective of the study was to describe the exposure to and the impact of the COVID-19 pandemic on important areas of life as well as related worries and sleep problems.

## Materials and methods

### Participants and study setting

The participants included in the current study were mothers of children participating in a double-blinded clinical trial (ClinicalTrials.gov Identifier: NCT02272842; Universal Trial Number: U1111-1161-5187) entitled "The effect of Vitamin B12 supplementation in Nepali Infants on Growth and Neurodevelopment" [24]. A total of 600 children of 6–11 months enrolled in the trial were followed every 12 months after a one-year supplementation period. From 9 September 2020 to 10 February 2021, we interviewed 493 of the mothers on COVID-19-related questions. The age of the children of the participating mothers were 54–71 months.

A total of 67 children from the original study of 600 children were lost to follow up (due to migration and refusal) when we initiated the COVID-19 exposure interviews with the mothers. Of the remaining 533 participants, we were not able to complete the COVID-19 exposure questionnaire with 40 mothers and thus the final sample consisted of 493 mothers. The flow of participants is depicted in **Fig 1**.

### Procedures

At enrollment, mothers answered questions on demographics and household characteristics. Information on the family members included mother's age, age category of family members, caste/ethnicity of family, level of education, occupation group, and nutritional status of the mother. Family members were categorized in age groups for which we considered children <16 and adults >60 more vulnerable to consequences of a COVID-19 infection. Caste/ethnicity of the families was categorized as Newar, Brahmin, Chhetri, Tamang and others. The educational level was assessed as illiterate, primary school, secondary school, School Leaving Certificate/intermediate school, bachelor's degree, and above. For our analysis, we dichotomized education up to $10^{th}$ years (i.e., completed secondary school) and above. For occupation, we categorized the occupation groups of both mothers and fathers as no formal work, agriculture, carpet worker, daily wage earner, self-employed, working in the service, and working abroad. Trained staffs measured height and weight in order to calculate body mass index (BMI) for the nutritional status of the mothers using a stadiometer for height (Prestige, HardikMedi Tech, India) and an electronic scale for weight (Salter/HoMedics Group, UK and Seca, Germany).

Socioeconomic status was measured by a multidimensional measure, the WAMI-index with the components; water and sanitation, household assets, maternal education, and income [25]. In contrast to previous studies, our data did not have income information and hence the index was calculated using the first three dimensions [26] (see S1 Table). For the current study, we set the cutoff for a low WAMI score at the $33^{rd}$ percentile of our study sample.

In follow-ups of the original study when the children were 30–35 months old, the mothers answered the Self-Reporting Questionnaire-20 (SRQ-20). The scores from SRQ-20 were from 2 years prior to the current study and covered previous maternal symptoms of depression.

The COVID-19 follow-up was completed during a visit to the study clinic when the children were in between 54 and 71 months old. Trained study staff administered the questionnaires with the mothers in a private room at the study clinic. Women who did not attend this follow-up visit for the present study (approximately 50%) were interviewed through phone calls due to restrictions of movement (lockdown).

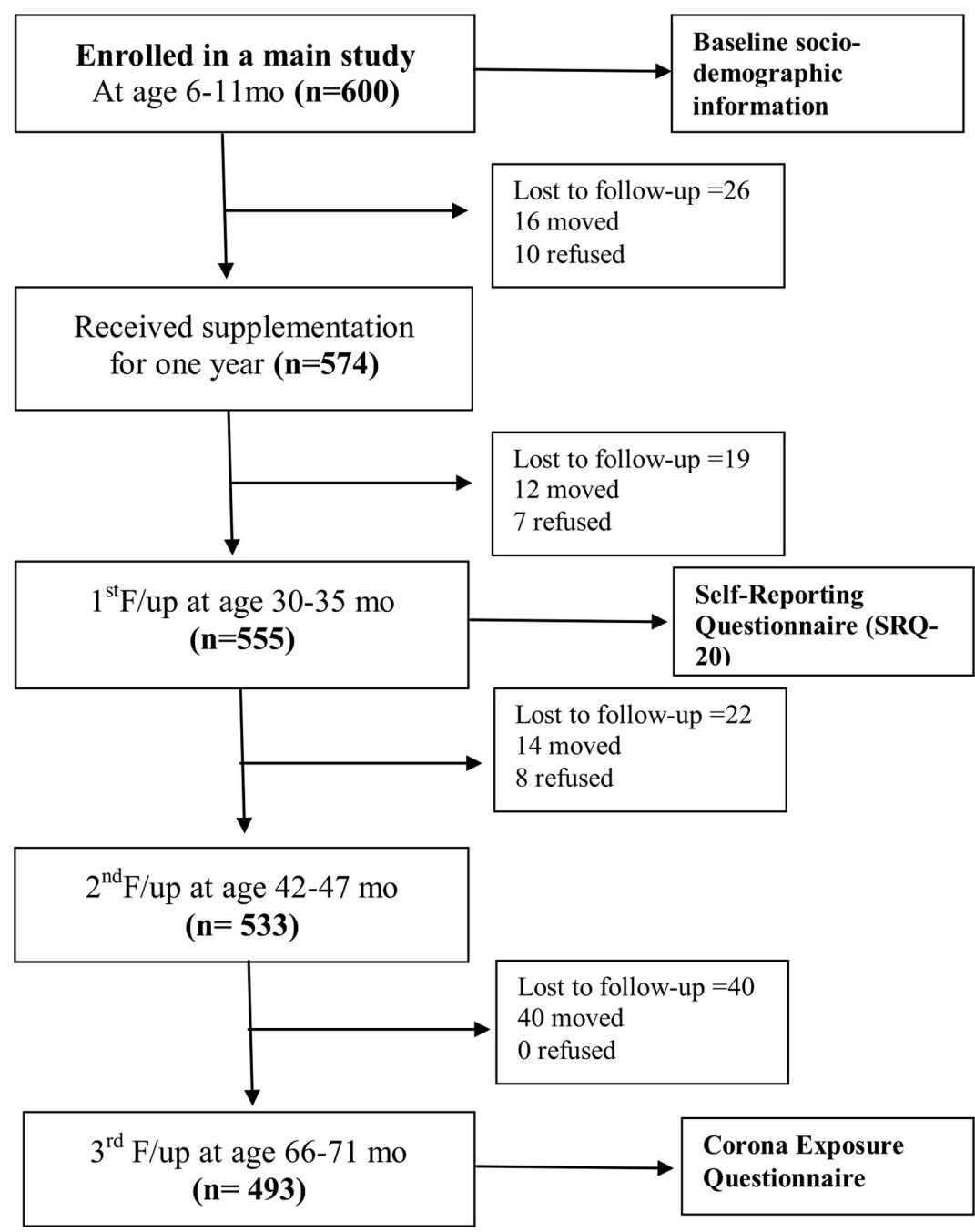

**Fig 1. Recruitment of mothers in Bhaktapur, Nepal.**

## Instruments

We assessed COVID-19 exposure through a questionnaire particularly developed for this study (See S1 File). The presence of symptoms during the last two weeks that had been described for COVID-19 was recorded by the major symptom categories (fever, cold/cough/difficulty breathing, sore throat and diarrhea). To identify whether the mother and her family members had a chronic illness or diseases that constituted a risk for severe COVID-19 illness,

a checklist with the following illnesses was included; chronic respiratory illness, renal disease, high blood pressure, diabetes, other chronic diseases. If the mother or another family member had such medical condition, we further asked the concerns about the shortage of essential medicines related to the illnesses. Furthermore, we recorded whether the mother had been exposed to someone who was COVID-19 positive, if she or anyone from her family had received COVID-19 positive test results, and how many days she previously had been quarantined or isolated. Prior to the study, our study team had thorough discussion of the items in which physicians and psychologists from Nepal and clinical psychologists from Norway were involved.

We also asked if the mothers believed that the life of someone close to her or her own life was in danger (response categories "not at all," "Some/Just a little," and "a great deal"). These items of the questionnaire are similar to previous traumatic exposure questionnaires [27, 28]. The impact of the pandemic on different aspects of life (i.e., economic, food security, employment, health related issues, daily life) was assessed on a five-point scale: "Not at all" (1); "a little" (2); "some" (3); "quite a lot" (4); and "a great deal" (5). This is a similar approach to what was used in the same region in relation to the impact of earthquakes [28].

Worry of contracting the corona virus was measured by two items, of which one was worry about contracting the corona virus yourself, and one was the worry that someone in the family would contract the corona virus. The worry items are based on a previous questionnaire on worry for infections [29]. Three items were on sleep related problems covering if the mothers slept worse than before due to worry about health-related, social and economic consequences due to the corona virus. Scoring for both worry and sleep problems was recorded on a five-point Likert scale (strongly disagree to strongly agree). The score ranges from 2 to 10 and 3 to 15 for worry and sleep respectively. The internal consistency of the worry and sleep score were excellent, with a Chronbach's Alpha for itemized data of 0.93 and 0.95 respectively.

The Self-Reporting Questionnaire-20 (SRQ-20) [30] is a brief screening questionnaire for mental health problems consisting of 20 items on symptoms of depression, anxiety and psychosomatic complaints. It can be administrated through interview or the paper-pencil method. The response from each item was recorded as Yes or No depending on whether the symptoms were present or not during the last 30 days. The scores are summed up to a total score that ranges from 0 to 20. It takes 5–10 minutes to administer. In the current study, we used 10 as a cutoff for depression [31]. The feasibility of the questionnaire has already been confirmed in a previous study within the same population [32].

The instruments were translated into Nepali by a psychologist and medical doctors and then back-translated by a qualified person independent from the study who was fluent in both Nepali and English language, following the standard guidelines for translation processes [33]. Discrepancies were discussed and adjusted.

## Ethics statement

Ethical approvals were obtained from Nepal Health and Research Council (NHRC; #73/2017, #820/2020) and from the Regional Committee for Medical and Health Research Ethics (REC; No.2014/1528) in Norway. Mothers were asked to visit the central clinic and we obtained informed written consent for the extended follow ups.

## Statistical analysis

All the continuous variables are presented in means and standard deviations, and categorical variables in numbers and percentages. Exposure variables were categorized into a three-point scale: "Not at all" (1 from original questionnaire), "Some" (merged 2 and 3 from original

questionnaire) and "A great deal" (merged 4 and 5 from original questionnaire). We compared the mean worry and sleep scores between groups using the student's t-test. The high risk groups were dichotomized as follows: education of mother and father (up to 10[th] years vs. above 10[th] years), symptoms similar to COVID-19 (absence vs. presence of at least one symptom), chronic disease of mother (absence vs. presence of at least one chronic disease), chronic disease in family member (absence vs. presence of at least one chronic disease), previous symptoms of depression (scores above 10 vs. 10 and under), and the WAMI-index score ($\leq$33[rd] percentile vs. >33[rd] percentile of the WAMI score). The missing values were handled by listwise deletion. Data was analyzed using the STATA 16.0 software.

## Results

The mean age of the mothers was 32.3 (SD: 4.6) years, more than 75% belonged to the Newar ethnic group, and approximately 50% resided in a joint family. Half of the families owned land, and 57% lived in their own house. Around 55% of both the mothers and fathers had an educational level up to 10 years (**Table 1**).

Among all mothers, 26 (5.4%) had either a positive COVID-19 case in the family or had been exposed to someone who had tested positive in the previous two weeks. Among these 26, the mean (SD) number of days of the quarantine was 15.7 (10.1) (**Table 2**).

Among all mothers, 188 participants, either herself or her family members, had chronic diseases and among these, 1.4% were concerned about the shortage of essential medicines due to the pandemic. Less than 10% believed to a great deal that their life or the life of someone close to them was in danger. Approximately 22% reported that the pandemic had to a great deal a negative effect on family life. More than 29% reported that they had economic problems, and 33% reported that their employment was greatly affected by the pandemic. Approximately 9% were to a great deal concerned that they would experience health related issues. About 16% of the mothers felt that food security and daily life were greatly affected by the pandemic (**Table 3**).

The mean (SD) worry and sleep score were 5.4 (2.5) and 4.9 (2.8), respectively. The distributions of all the response categories for each item of worry and sleep are shown in Table 4. Among the mothers, 37.1% reported worry related to contracting the corona virus and 39.5% reported worry that someone in their family would contract the virus. Similarly, 7.5% slept worse due to worry about health-related consequences of the corona virus, 7.1% due to social consequences, and 8.9% due to economic consequences.

Table 5 shows the difference in worry and sleep scores based on high- vs. low-risk groups. Mothers and fathers who had more than 10 years of education and high WAMI index score had higher worry scores than those with 10 years of education and with a low WAMI index score. There were no significant differences in the worry score according to the other risk groups. The only risk-group with a significantly higher mean score for sleep problems was mothers who had a previous history of symptoms of depression. There were significant differences in the worry score according to the mode of data collection (in-person vs, phone call) (See S2 Table).

## Discussion

The present study revealed a considerable impact on the everyday life of the Nepalese mothers in relation to the COVID pandemic between 9 September 2020 and 10 February 2021. Approximately one third of the mothers were heavily affected by the COVID-19 pandemic in terms of their economy and employment, and for 16% their daily life and food security were perceived as severely affected. Only a small group reported major impact on their health.

**Table 1. Socio-demographic and clinical characteristics of 493 mothers in Bhaktapur, Nepal.**

| Characteristics | Number N or mean | Proportion% or SD |
|---|---|---|
| **Demographic Characteristics:** | | |
| Age of mother (years), m (sd) | 32.3 | 4.6 |
| **Caste/Ethnicity of mother** | | |
| Newar | 374 | 75.8 |
| Brahmin | 15 | 3.0 |
| Chhetri | 15 | 3.0 |
| Tamang | 64 | 12.9 |
| Others | 25 | 5.1 |
| **Age category of family members** | | |
| <1year | 75 | 15.2 |
| 1–16 years | 493 | 100.0 |
| Senior citizens (>60 year) | 207 | 41.9 |
| **Socio-economic status:** | | |
| Family staying in joint family | 234 | 47.4 |
| Family having own land | 253 | 51.3 |
| Family living in own house | 281 | 57.0 |
| Remittance from abroad | 47 | 9.5 |
| Number of rooms used ($\leq 2$) | 259 | 52.5 |
| WAMI Index, m (sd) | 0.6 | 0.1 |
| **Mother's education** | | |
| Up to 10th years | 267 | 54.2 |
| Above 10th years | 226 | 45.8 |
| **Father's education** | | |
| Up to 10th years | 275 | 55.8 |
| Above 10th years | 218 | 44.2 |
| **Mother's occupation** | | |
| No formal work | 285 | 57.8 |
| Agriculture | 22 | 4.5 |
| Carpet worker | 12 | 2.4 |
| Daily wage earner | 57 | 11.6 |
| Self-employed | 62 | 12.6 |
| Services | 54 | 10.9 |
| Working abroad | 1 | 0.2 |
| **Father's occupation** | | |
| No formal work | 15 | 3.0 |
| Agriculture | 14 | 2.8 |
| Carpet worker | 4 | 0.8 |
| Daily wage earner | 192 | 38.9 |
| Self-employed | 146 | 29.6 |
| Services | 96 | 19.5 |
| Working abroad | 26 | 5.3 |
| **Nutritional Status of the mother at baseline:** | | |
| Height, m(sd) | 150.0 | 5.3 |
| Weight, m(sd) | 53.9 | 8.6 |
| BMI, m(sd) | 23.9 | 3.6 |

**Table 2. COVID-19 exposure in 493 mothers and their family in Bhaktapur, Nepal.**

| Characteristics | Yes N | Proportion % |
|---|---|---|
| **Symptoms within the last 2 weeks:** | | |
| Fever | 46 | 9.3 |
| Cough/Cold/Difficulty breathing | 88 | 17.8 |
| Sore Throat | 49 | 9.9 |
| Diarrhea | 14 | 2.8 |
| **Health condition of the mother (n = 30\*):** | | |
| Chronic respiratory illness | 5 | 1.0 |
| Renal diseases | 2 | 0.4 |
| High blood pressure | 15 | 3.0 |
| Diabetes | 8 | 1.6 |
| Other chronic illnesses | 9 | 1.8 |
| **Health condition of family members (n = 174\*)** | | |
| Chronic respiratory illness | 44 | 8.9 |
| Renal diseases | 6 | 1.2 |
| High blood pressure | 122 | 24.7 |
| Diabetes | 73 | 14.8 |
| Other chronic illnesses | 38 | 7.7 |
| **Corona virus exposure** | | |
| Exposure to positive cases | 26 | 5.3 |
| Positive cases in family | 26 | 5.3 |
| Number of days of quarantine (n = 18), m (sd) | 15.7 | 10.1 |

\* One or more chronic diseases.

Worry about contracting the virus showed a social gradient, with more worry in the higher socioeconomic status groups. Sleep problems related worry was not common, but higher for those with a previous history of symptoms of depression.

## Negative impacts of COVID-19 on daily life

The mothers reported that the economic and employment status were greatly affected by the pandemic. At baseline for the original study, many reported to have no formal work, being in agricultural work, carpet work and being a daily wage earner and about half of their husbands had the same occupations signifying that the type of work in Bhaktapur is mainly based on

**Table 3. Negative impact of COVID-19 in 493 mothers in Bhaktapur, Nepal.**

| | Not at all | | Some | | A great deal | |
|---|---|---|---|---|---|---|
| | N | % | N | % | N | % |
| Concerns about shortage of essential medicines | 409 | 82.9 | 77 | 15.6 | 7 | 1.4 |
| Believe that own life or the life of someone close to mother was in danger | 307 | 62.3 | 140 | 28.4 | 46 | 9.3 |
| Negative effect on family life | 90 | 18.3 | 296 | 60 | 107 | 21.7 |
| Which aspect of life has the pandemic affected you? | | | | | | |
| Economic | 122 | 24.8 | 226 | 45.8 | 145 | 29.4 |
| Food security | 221 | 44.8 | 193 | 39.2 | 79 | 16 |
| Employment | 118 | 23.9 | 211 | 42.8 | 164 | 33.3 |
| Health related issues | 263 | 53.4 | 188 | 38.1 | 42 | 8.5 |
| Daily life | 168 | 34.1 | 243 | 49.3 | 82 | 16.6 |

**Table 4. Response characteristics of worry and sleep items in Nepalese mothers (n = 493).**

| | Mean | SD | Range | Alpha[1] | |
|---|---|---|---|---|---|
| **Total Worry score** | **5.4** | **2.5** | **2–10** | **0.93** | |
| **Total Sleep score** | **4.9** | **2.8** | **3–15** | **0.95** | |
| **Items** | | Prevalence | | | |
| | **Strongly Disagree** | **Disagree** | **Neither Disagree nor Agree** | **Agree** | **Strongly Agree** |
| 1. I worry that I will contract the corona virus. | 28.6 | 23.3 | 10.9 | 31.2 | 5.9 |
| 2. I worry that someone in my family will contract the corona virus. | 24.7 | 21.5 | 14.2 | 32.8 | 6.7 |
| 3. I sleep worse than before due to worry regarding the health consequences of the Corona virus. | 57 | 31.8 | 3.6 | 4.9 | 2.6 |
| 4. I sleep worse than before due to the social consequences of the Corona virus | 59.6 | 29.8 | 3.4 | 4.5 | 2.6 |
| 5.I sleep worse than before due to worry regarding the economical consequences of the Corona virus. | 57.8 | 30.0 | 3.2 | 5.7 | 3.2 |

[1] Chronbach's alpha.

daily wage earnings. These were occupations that were almost impossible to keep functioning during lock down or other restrictions. Other aspects of the daily life, like food insecurity, health related issues, shortage of essential medicines were also marginally prevalent. These results are in line with a previous review of published articles related to the psychosocial impact of COVID-19 in Nepal showing substantial impact on economy, agriculture, employment and the health sector [4]. We found a similar pattern of responses in the same study sample during the massive earthquakes in Nepal in 2015, where the most affected aspect of daily

**Table 5. Mean worry and sleep scores according to the exposure, chronic illness, previous depression, and socioeconomic status of Nepalese mothers (n = 493).**

| | | Worry Score | | Sleep Score | |
|---|---|---|---|---|---|
| **Variables** | **N** | **Mean (SD)** | **p-value** | **Mean (SD)** | **p-value** |
| **Symptoms similar to COVID-19** | | | | | |
| Absent | 365 | 5.31 (2.63) | 0.343 | 4.80 (2.70) | 0.131 |
| Present (At least one) | 128 | 5.56 (2.37) | | 5.23 (2.98) | |
| **Chronic Disease of mother or other family member** | | | | | |
| Absent | 305 | 5.34 (2.56) | 0.689 | 4.86 (2.73) | 0.594 |
| Present (At least one) | 188 | 5.43 (2.57) | | 5 (2.86) | |
| **Exposed to positive case** | | | | | |
| No | 467 | 5.15 (2.63) | 0.648 | 4.89 (2.77) | 0.417 |
| Yes | 26 | 5.39 (2.56) | | 5.34 (3.00) | |
| **Previous maternal depression** | | | | | |
| Score < 10 | 422 | 5.36 (2.58) | 0.719 | 4.79 (2.68) | 0.020* |
| Score ≥ 10 | 71 | 5.48 (2.50) | | 5.62 (3.27) | |
| **Mother's education** | | | | | |
| Up to 10 years | 267 | 5.13 (2.60) | 0.020* | 4.94 (2.78) | 0.801 |
| Above 10 years | 226 | 5.67 (2.49) | | 4.88 (2.79) | |
| **Father's education** | | | | | |
| Up to 10 years | 275 | 5.09 (2.55) | 0.005* | 4.91 (2.86) | 0.985 |
| Above 10 years | 218 | 5.73 (2.54) | | 4.91 (2.68) | |
| **WAMI Index** | | | | | |
| ≤ 33percentile | 143 | 4.98 (2.60) | 0.030* | 4.91 (2.88) | 0.99 |
| >33 percentile | 350 | 5.53 (2.54) | | 4.91 (2.74) | |

life was employment, and less on health-related issues. However, in comparison to the present results, the perceived impact on food insecurity was higher while economic impact was almost not affected [28].

Although many mothers reported that they or someone in their family had a chronic disease, the mothers were not so much concerned about supplies of essential medicines. This could be related to the fact that the study setting is close to the capital city of Nepal where the government made more efforts to prevent the pandemic compared to other regions of the country. In another study from Eastern less urban areas of Nepal, people perceived that medical supply was the most affected aspect of the health care system in Nepal during the pandemic [4].

## Negative impacts of COVID-19 on worry and sleep

Around 10% of the participants believed that their own life or the lives of someone close to them were in danger. Approximately 40% of the participants reported being worried that they themselves or their family members would contract the corona virus. A high rate of worry is expected during a pandemic and in line with previous studies that demonstrated an increase in public anxiety during epidemics or pandemics such as the 2003 SARS pandemic, the 2009/2010 H1N1 pandemic and the 2014/2016 Ebola pandemic [34–37]. Worry about the COVID-19 pandemic can also induce negative effects on mental health in the general public [38].

The rate of worry related sleep problems were low in the present study. Despite the major impact of the pandemic on their daily life, the mothers reported that the social, economic and health-related consequences due to the COVID-19 pandemic did not negatively affect their sleep and less than ten percent of the mothers reported these problems. We do not have any possible explanation to the low prevalence of sleep problems since we lack specific information about the sleep patterns of Nepalese women. However, the low prevalence might be due to taking COVID-19 very lightly since most of the people who were infected during the study period were asymptomatic or had mild symptoms.

The mothers who had a previous history of depression reported more sleep problems due to concerns with the COVID-19 pandemic. This finding is in accordance with previous studies showing that depressive symptoms is associated with poorer or disrupted quality of sleep during the pandemic [17, 39], and that previous history of psychological condition such as anxiety, stress and depression is associated with poor sleep [40].

There was a social gradient in worry, but in the opposite direction than expected from previous studies in other parts of the world. In the present study, there were higher levels of worry for contracting the virus among those who had an educational level higher than secondary. This group with the lowest educational qualifications being least concerned about contracting COVID-19, might be the challenging factor in controlling the pandemic due to less safety measures to protect them from virus infection. Hence, public health practitioners should consider the level of education while addressing the psychological impacts of pandemics since our results suggests that education can be the deciding factor for psychological impact of the individuals. Further, and also in contrast to our expectations and the established social gradient of health that has been evident during the pandemic [41], we found that those with higher SES (i.e. higher WAMI index) had higher worry scores than those with lower SES. This pattern of higher level of worry and higher socioeconomic status have been reported previously in high risk communities of Africa with higher academic qualification associated with greater level of psychological distress and worry [42, 43]. This may be related to more access to information about the pandemic and consequences for those with higher education [44]. Health literacy is an individual's idea, motivation and capacities to recognize, evaluate and judge health

information [45]. Individuals with low SES may be less sensitive toward health-related and other consequences of the pandemic due to the lack of health literacy resulting in these being less likely to seek help when they experienced symptoms [46]. Secondly, they might lack the capability of understanding the symptoms and health consequences of the corona virus limiting their interpretation and decision about own health condition [47].Thus, belonging to a low SES family may lead to less worry scores among mothers in our study. Moreover, all people who were infected from the corona virus during the study period, were asymptomatic or had mild symptoms [48]. This may have led to a misinterpretation of the disease in people with low SES.

One in every three of the mothers reported at least one of the major chronic diseases in herself or in someone in her family. Some of these common conditions such as respiratory diseases, diabetes, and chronic kidney diseases, have been related to increased risk of deaths from COVID-19 in Nepal [49]. Still there were no differences in the worry and sleep related worry scores between the participants who had these conditions or with family members with these conditions. The lack of associations with worry might be due to the younger age group and the predominance of the less hazardous 1st variant of SARS-Cov-2 in Nepal at the time of the data collection [48].

## Strengths and limitations

A major strength of our study is the large sample size in a well-maintained cohort of mothers of young children. A limitation is that due to lockdowns, approximately half of the questionnaires were collected through phone calls which might not be as effective as interviews through direct contact. We also found that the worry score was significantly higher when collecting by an in-person interview compared to through a phone call. The fact that people respond differently if they were interviewed through phone calls rather than through direct contact methods have been demonstrated before [50].The study population was also limited to mothers whose infants were enrolled in a randomized controlled trial from 2015 to 2017 which limits the generalizability of the results since the sample may differ from general populations. Although the population of the Bhaktapur district covers different migrants from the whole areas of Nepal, the results can only be generalizable to the urban settings of Nepal with similar characteristics such as availability of daily supplies and health facilities, and the restrictions to the pandemic casted by local governments. We have only included mothers in the present study, and whether the results can be generalized to fathers is uncertain. However, since women often have responsibilities for health and care and may also be important role models for children, their perceptions and attitudes may also impact the family's preventive measures and health. Our data may present only the results from a certain period of the pandemic in Nepal and hence cannot be generalized to the pandemic as a whole. The questionnaire was brief due to the situations for the mothers, and other aspects that would have been of interest such as more detailed information on sleep and current mental health could not be measured due to time constraints. The measure of depression is limited to a self-report questionnaire that we collected 2 years before, and not by a clinical interview. The final sample was better off than those who were lost to follow up. The result can be the strength of our study to differentiate the SES indicating that the losses to follow up were mainly due to the difficult financial situation during the lockdown (See S3 Table).

## Conclusions

Our study reveals a considerable negative impact of the COVID-19 pandemic on everyday life of mothers across major domains such as economy and employment, food security and

everyday life in relation to the restrictions. Mothers with high socioeconomic status were more worried that they might contract the virus.

## Supporting information

**S1 File. Corona virus exposure questionnaire.**
(DOCX)

**S1 Table. Calculation of the WAMI index.**
(DOCX)

**S2 Table. Mean Worry and Sleep scores according to the mode of data collection of Nepalese mothers.**
(DOCX)

**S3 Table. Age and socioeconomic factors according in those that were lost to follow-up compared to those who were included in the study sample.**
(DOCX)

## Acknowledgments

We would like to show appreciation to all the field staff and mothers who participated in this study. We are also thankful to the Child Health Research Project Team at the Department of Child Health, Institute of Medicine, Tribhuvan University and Siddhi Memorial Foundation.

## Author Contributions

**Conceptualization:** Suman Ranjitkar, Tor A. Strand, Manjeswori Ulak, Ingrid Kvestad, Merina Shrestha, Ram K. Chandyo, Laxman Shrestha, Mari Hysing.

**Data curation:** Suman Ranjitkar, Tor A. Strand, Ram K. Chandyo.

**Formal analysis:** Suman Ranjitkar, Tor A. Strand.

**Funding acquisition:** Tor A. Strand, Ingrid Kvestad, Ram K. Chandyo, Laxman Shrestha, Mari Hysing.

**Investigation:** Suman Ranjitkar, Manjeswori Ulak, Ram K. Chandyo.

**Methodology:** Suman Ranjitkar, Ingrid Kvestad, Ram K. Chandyo, Mari Hysing.

**Project administration:** Suman Ranjitkar, Manjeswori Ulak, Ram K. Chandyo.

**Supervision:** Ram K. Chandyo, Mari Hysing.

**Writing – original draft:** Suman Ranjitkar, Tor A. Strand, Ingrid Kvestad, Mari Hysing.

**Writing – review & editing:** Suman Ranjitkar, Tor A. Strand, Manjeswori Ulak, Ingrid Kvestad, Merina Shrestha, Catherine Schwinger, Ram K. Chandyo, Laxman Shrestha, Mari Hysing.

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
