## [Decision Letter · Decision Letter 0]

13 Dec 2021

PGPH-D-21-00572

Impact of the COVID-19 pandemic on daily life among mothers in Bhaktapur, Nepal

Dear Dr. Ranjitkar,

Thank you for submitting your manuscript to PLOS Global Public Health. After careful consideration, we feel that it has merit but does not fully meet PLOS Global Public Health’s publication criteria as it currently stands. Therefore, we invite you to submit a revised version of the manuscript that addresses the points raised during the review process.

We look forward to receiving your revised manuscript.

Kind regards,

Rachel Hall-Clifford

Academic Editor

Journal Requirements:

3. Please ensure you have included the registration number for the clinical trial referenced in the manuscript.

4. Please provide separate figure files in .tif or .eps format only, and remove any figures embedded in your manuscript file.

5. Please update the completed 'Competing Interests' statement, including any COIs declared by your co-authors. If you have no competing interests to declare, please state "The authors have declared that no competing interests exist". Otherwise please declare all competing interests beginning with the statement "I have read the journal's policy and the authors of this manuscript have the following competing interests:

6. In the online submission form, you indicated that [Insert text from online submission form here]. All PLOS journals now require all data underlying the findings described in their manuscript to be freely available to other researchers, either 1. In a public repository, 2. Within the manuscript itself, or 3. Uploaded as supplementary information.

7. Please amend your] detailed Financial Disclosure statement. This is published with the article, therefore should be completed in full sentences and contain the exact wording you wish to be published.

i) Please include all sources of funding (financial or material support) for your study. List the grants (with grant number) or organizations (with url) that supported your study, including funding received from your institution. 

ii). State the initials, alongside each funding source, of each author to receive each grant.

iii). State what role the funders took in the study. If the funders had no role in your study, please state: “The funders had no role in study design, data collection and analysis, decision to publish, or preparation of the manuscript.”

Additional Editor Comments (if provided):

This is an important contribution to our understanding of the COVID-19 pandemic on daily life and well-being. In addition to the suggestions of the reviewers, I would add:

1) Since your framing of the paper and discussion focus on worry, please title the paper accordingly. This is a very interesting framing, and the title should reflect that. Please further incorporate in the introduction that you are considering "worry" not just about COVID-19 itself but many of its sequelae in Nepal (i.e., financial outcomes, loss of childcare, etc.).

2) Please indicate the age range of children in the abstract. This is very clear in the methods, but after reading the abstract, I thought certainly outcomes would vary by age of children (which turned out to not be relevant as I saw the study design).

3) In the discussion, please include more thinking about the gendered experience of women.

Thank you for this submission!

Reviewers' comments:

Reviewer's Responses to Questions

**Comments to the Author**

1. Does this manuscript meet PLOS Global Public Health’s publication criteria? Is the manuscript technically sound, and do the data support the conclusions? The manuscript must describe methodologically and ethically rigorous research with conclusions that are appropriately drawn based on the data presented.

Reviewer #1: Yes

Reviewer #2: Yes

2. Has the statistical analysis been performed appropriately and rigorously?

Reviewer #1: Yes

Reviewer #2: Yes

3. Have the authors made all data underlying the findings in their manuscript fully available (please refer to the Data Availability Statement at the start of the manuscript PDF file)?

Reviewer #1: Yes

Reviewer #2: Yes

4. Is the manuscript presented in an intelligible fashion and written in standard English?

Reviewer #1: Yes

Reviewer #2: No

5. Review Comments to the Author

Reviewer #1: This study uses cross-sectional survey data to assess the impact of the COVID-19 pandemic on various aspects of daily life for mothers in Bhaktapur, Nepal. While previous studies in Nepal have assessed psychological impacts during the early pandemic period, this study centers on a later phase of the pandemic (September 2020–February 2021, prior to the predominance of the SARS-CoV-2 Delta variant). Findings of higher pandemic-associated worry in mothers with higher educational attainment and socioeconomic status are concordant with a prior, national-level study assessing pandemic distress in April-May 2020 (Shrestha et al. 2020). Readers of PLOS Global Public Health may be interested in implications of these findings for health education efforts in Nepal, as well as suggestions for addressing these psychological impacts.

Major and minor comments for the authors are provided below.

INTRODUCTION

The authors provide a comprehensive overview of previous literature on the impact of COVID-19 on psychological health in general and in Nepal specifically.

Additional contextual information about Bhaktapur municipality (e.g. population demographics, geographical location within Nepal) would be appreciated in the Introduction, as well as in the Discussion of the study findings.

METHODS

The authors have clearly described the participant recruitment strategy and reasons for loss to follow up.

Definition of variables is easily understood, but what is meant by “age category of family members” is not immediately clear.

The methods for statistical analysis are technically sound, although authors should discuss explicitly their strategy for handling any missing data.

The authors received ethical approvals from the Nepal Health and Research Council (NHRC) and Regional Committee for Medical and Health Research Ethics (REC) and described obtaining written consent from study participants. Authors have made data available upon request in accordance with requirements of NHRC and REC.

RESULTS

Results are communicated clearly.

Did the authors note any difference in key findings (negative impacts of COVID-19) by mode of data collection (phone vs. in-person interview)?

TABLES

Tables 1-5 and Supplemental Table 1 are well-organized and easily comprehensible.

DISCUSSION/CONCLUSIONS

The authors appropriately situate their findings within a broader discussion of previous literature on pandemic distress in Nepal and in other LMICs.

Authors have thoughtfully considered the Strengths and Limitations of the study, including generalizability, variability in the mode of data collection (phone and in-person interviews), and inability to collect data on depressive symptoms through clinical interview. Authors should additionally address how the study setting in Bhaktapur affects generalizability.

Authors should expound on implications of their findings for readers of PLOS Global Public Health. What are the implications for health education efforts in Nepal if those least concerned about COVID-19 are those who with the lowest educational attainment? What should public health practitioners consider when aiming to address the psychological impacts the findings highlight?

FIGURES

Figure 1 is easily comprehensible. The title should be reworded to reflect the participants of the current study, e.g. “Recruitment of mothers in Bhaktapur, Nepal”.

MINOR COMMENTS:

Use of the term “Corona” should be replaced with “Coronavirus” or “COVID-19,” including in the title of Table 2.

This article satisfies the below criteria for acceptance to PLOS Global Public Health:

1. The study presents the results of primary scientific research.

2. Results reported have not been published elsewhere.

3. Experiments, statistics, and other analyses are performed to a high technical standard and are described in sufficient detail.

4. Conclusions are presented in an appropriate fashion and are supported by the data.

5. The article is presented in an intelligible fashion and is written in standard English.

6. The research meets all applicable standards for the ethics of experimentation and research integrity.

7. The article adheres to appropriate reporting guidelines and community standards for data availability.

Reviewer #2: The authors present findings of a study that assessed effect of COVID-19 in an urban district of Nepal. The article is largely coherent however it would benefit with some specific detail in few places. I would strongly recommend the article to be professionally edited for language as there are typos and grammatical errors in the article (e.g., line 90, 207, 255-256).

Line number Comments

103 Suggest to include some information on characteristics of respondents who were lost to follow up. Was it different from the final sample?

106 Need to specific and consistent in using the term. Caregivers or mothers. Grandparents, father or other members of an extended family can also be a caregiver

113 What was the definition for” no work” specially for mothers (57.8%)? Did it also include housewife or household work? Strongly suggest to use alternative word

135 Recommend including information tool development of Corona Exposure questionnaire. Was this questionnaire pre-tested?

213-215 Convoluted sentence. Please make this sentence easier to read

269 What could be the possible explanation about the low prevalence of sleep problem – What do authors think about that? - more resilience because of repeated stressors? or taking COVID-19 lightly?

287-289 Argument here is really not clear. Please add more reasoning about why they could be less sensitive

6. PLOS authors have the option to publish the peer review history of their article (what does this mean?). If published, this will include your full peer review and any attached files.

**Do you want your identity to be public for this peer review?** For information about this choice, including consent withdrawal, please see our Privacy Policy.

Reviewer #1: No

Reviewer #2: No

---

## [Editor Report · Decision Letter 1]

2 Mar 2022

Impact of the COVID-19 pandemic on daily life and worry among mothers in Bhaktapur, Nepal

PGPH-D-21-00572R1

Dear Mr. Ranjitkar,

We are pleased to inform you that your manuscript 'Impact of the COVID-19 pandemic on daily life and worry among mothers in Bhaktapur, Nepal' has been provisionally accepted for publication in PLOS Global Public Health.

Best regards,

Rachel Hall-Clifford

Academic Editor